# Successful outcomes with low–threshold intervention for cannabis use disorders in Norway - an observational study

**John-Kåre Vederhus** [1] *, **Malin Rørendal**[2], **Madelene Skårdal**[3], **Marianne Otterstad Næss**[4], **Thomas Clausen**[1], **Øistein Kristensen**[1]

**1** Addiction Unit, Sørlandet Hospital HF, Kristiansand, Norway, **2** City Centre Outreach Service, Social and Welfare Services, Oslo, Norway, **3** Social and Welfare Services, Kristiansand Municipality, Kristiansand, Norway, **4** Social and Welfare Services, Fredrikstad Municipality, Fredrikstad, Norway

* john-kare.vederhus@sshf.no

**Data Availability Statement:** An anonymized data set necessary to facilitate replication of the study

## Abstract

### Aims

Cannabis is the most commonly used regulated drug by European youths. Yet, few cannabis-specific interventions have been examined in Europe. The Cannabis Cessation Program (CCP) was developed in Sweden in the 1990s and has been implemented in some Norwegian municipalities. The present study aimed to examine outcomes of this intervention in the Norwegian setting.

### Method

The respondents (N = 102) were recruited in four community-based CCPs in Norway. We examined their changes in cannabis use, other substance use, mental distress, well-being, sense of coherence (SoC), and social networks, from baseline (T0) to post-treatment (T1) and up to a 3-month follow-up period (T2). Changes were evaluated with pair-wise t-tests.

### Result

Seventy-six participants (75%) completed the 8-week program, according to plan. All participants reported a significant reduction in cannabis use at T1 (average reduction ~16 days per month) and at T2 (N = 59; ~13 days per month). Among those that completed the program, 67% was abstinent from cannabis at T1 and 37% was abstinent at T2. An intention-to-treat analysis showed that 50% (51/102) and 22% (22/102) were abstinent from cannabis use at T1 and T2, respectively. In parallel to abstinence, we observed a substantial reduction in mental distress and an increase in well-being and SoC. Respondents socialized with fewer friends with current substance use, but drug-free social networks were not expanded.

### Conclusion

Our findings suggested that the CCP was a valuable, low-threshold manual-based intervention for cannabis use disorders. It showed considerable potential for reducing individuals' cannabis use.

findings has been uploaded as a Supporting Information file.

**Funding:** The authors received no funding for this work.

**Competing interests:** The authors declare that no competing interests exist.

## Clinical trial registration

Clinicaltrials.gov no. NCT04989205. Registered 12 July 2021, i.e., the study was retrospectively registered.

## Background

Cannabis is the most commonly used regulated drug in Europe–the prevalence of cannabis use is roughly five-fold higher than the prevalence of other drug use [1]. The latest estimate showed that 15.8 million young adults, aged 15–34 years, used cannabis in the last year, which comprised 15.4% of this age group [1]. An estimated 9% of individuals that use cannabis will develop a cannabis use disorder (CUD) [2]. CUD ranges from mild to severe, with signs of continued and even compulsive use, despite negative consequences [3]. The European Monitoring Center for Drugs and Drug Addiction (EMCDDA) has expressed a growing concern about the availability of high-potency products: cannabis resin sold in Europe has an average tetrahydrocannabinol content of 20% to 28%, almost twice that of herbal cannabis [1]. This availability may partly explain the increase in the proportion of patients with cannabis-related problems that enter the general substance use disorder (SUD) treatment system at the specialized level. For example in Norway, the number of patients in specialist health care that used cannabis as the primary drug increased by 40% from 2009 to 2015 [4, 5].

Cannabis-specific interventions are rare in the European treatment system, especially at the municipality level. In a recent Cochrane review on psychosocial interventions for CUD, only four out of 23 controlled studies were undertaken in Europe [6]. In the Nordic Council of Ministers' social and health sector, the Nordic Welfare Center has emphasized that services for young adults should have a strong local anchor in the municipalities, and local models of good evidence-based practices should be developed and utilized across the Nordic countries [7]. An encouraging example is Lundqvist and Ericsson's 'Cannabis Cessation Program' (CCP, Nordic abbreviation: HAP), developed in the 1990s, which has spread and is currently used in many Swedish municipalities [7, 8]. The model has mainly been used as an individual-based intervention [9], but it has also been tested in a group format and in online formats [10–12].

The recent Cochrane review showed that approximately 70% of intervention participants in controlled studies completed treatment as intended, and that participants that received a psychosocial intervention for CUD had favorable outcomes (i.e., fewer days of cannabis use) at follow-up, compared to those in inactive control conditions [6]. The most consistent positive outcomes were achieved with cognitive-behavioral therapy and motivational enhancement therapy. For cannabis-related problems, similar to other substance use and mental health problems, the outcomes of interventions remained unclear. However, the review concluded that none of the studies showed that the cannabis-specific intervention achieved superior outcomes to non-specific interventions [6].

In addition to controlled studies, observational studies are important, particularly for interventions that have not been examined in a randomized controlled study. Observational studies provide benefit in studies that involve a heterogeneous population with complex conditions, like SUD, and they can contribute clinical observations of participants in naturalistic settings [13]. The present observational study examined outcomes of young adults that participated in a relatively new intervention in Norway: the CCP. A previous pilot study showed that an encouraging 79% of those that completed the program (26 of 33 individuals) reported at least a

30-day cannabis abstinence period at the end of the intervention [12]. The present study represents a more formal evaluation of this intervention.

In the present study, we evaluated a low-threshold, community-based intervention for individuals that perceived their cannabis use as problematic. We examined changes in cannabis use, other substance use, mental distress, well-being, sense of coherence (SoC), and social network, from baseline to post-treatment, and from baseline to a 3-month follow-up. We also examined whether severity factors (i.e., substance use severity, mental distress) and motivation at baseline could predict abstinence from cannabis use post-treatment.

## Methods

### CCP in Norway

The CCP is a manual-based program that applies a combination of cognitive therapy, motivational interviewing, and psychoeducation. The CCP manual has been translated into eight languages, including Norwegian [14]. Since 2005, the CCP has been implemented as a low-threshold, community-based program in several Norwegian municipalities. In Norway, the CCP has been nicknamed "Ut av tåka" ("Out of the fog"), an expression that originated among the participants. The groups that first implemented the model in Norway (the Social and Welfare Services in Kristiansand and the City Center Outreach Service in Oslo) have continued with internal and external training. In addition, they have participated in establishing an educational program for CUD treatment at the University of Agder.

Participants began the program during the first period (up to 8 weeks) of cannabis use cessation. The program comprised ~15 individual, out-patient sessions that targeted three phases, termed: a medical (withdrawal) phase, a psychological phase, and a social phase. Each participant was shown how to reflect upon thinking patterns and behavioral patterns that were developed during cannabis use. This reflection was intended to prepare participants in facing common challenges related to each phase; e.g., the physical and psychological withdrawal problems related to quitting cannabis use (Table 1). The overarching aim of the program was to assist individuals in reorganizing the thinking patterns that they developed when using cannabis, and to increase the social and psychological coping skills that the participants needed to quit cannabis use. One session included a family meeting, for the youngest participants (<20 years old) and their parents.

### Participants and study procedures

The participants were recruited in four municipality-based CCP centers in Norway (Kristiansand, Oslo, Fredrikstad, and Bergen), from January 2013 to December 2016. The target group

**Table 1. Phases of the CCP-program[a].**

| |
|---|
| • **Phase one (weeks 1–2)** prepares the participant for the physical withdrawal symptoms that are commonly experienced when quitting cannabis; e.g., sleep difficulties, irritation, strange dreams/nightmares, sweating, poor appetite, and restlessness. No pharmacological intervention was utilized in any part of the study. |
| • **Phase two (weeks 2–3)** focuses on psychological problems, characterized by emotional ups and downs, anxiety, irritation, feelings of isolation, and loneliness. The therapist helps the user to analyze, verbalize, and reflect on emotions. |
| • **Phase three (weeks 3–8)** focuses on working with the participant's identity, social network, and plan forward; e.g., relapse prevention measures. The therapist helps the user develop coping strategies, instead of using cannabis, to deal with challenges in life. |

[a]Adapted from Lundqvist & Ericsson [8]

comprised individuals that had engaged in regular or daily cannabis use and were motivated to stop cannabis use. The formal inclusion criteria were a Severity of Dependence Scale (SDS) score ≥4 and age ≥16 years old. Exclusion criteria were polydrug use, where cannabis was not the predominating substance of use, and any psychiatric comorbidity that was considered too severe to handle at a community-based center. Some participants came into contact with the centers through support services in the cities (e.g., healthcare services, social security and welfare services, child welfare services, and school healthcare services), but no formal referral was required. Many participants obtained information through friends, other users, or the internet, and then they contacted the services. Thus, participation in the intervention was based on self-selection. Participation was free of charge. A pre-session included information about the CCP and the study. All decisions to use the intervention were made as part of routine clinical care; e.g., the study inclusion and exclusion criteria were similar to those applied in routine clinical care. All participants that provided informed written consent were assessed with the study questionnaires (see below). The intervention was conducted by therapists (e.g., psychologists, nurses, social workers) that were well experienced with the method.

Participants were recommended to set a date for quitting cannabis use. They were advised to reduce or quit cannabis use before starting the program, or during the first part of the program. Data were collected at baseline (T0), at the end of the ~ 8-week treatment period (T1), and at a 3-month follow-up (T2; i.e., ~ 5 months after baseline).

The present study had a one-sample, pre- and post-test design. The aim was to examine the outcomes of the program.

## Measures

The inventory included basic socio-demographic variables. The SDS was used to assess the perceived severity of cannabis use [15]. Four items on the SDS investigated the respondent's concerns about cannabis use (e.g., "Do you think your use of cannabis is out of control?"), and one item asked how difficult it would be to stop or go without cannabis use. The items were scored on a 4-point scale. For example, concerns were scored as: 0 = never/almost never; 1 = sometimes; 2 = often; 3 = always/nearly always); and difficulty was scored as: 0 = not difficult; 1 = quite difficult; 2 = very difficult; 3 = impossible. The sum score ranged from 0–15, with higher scores indicating higher levels of severity. The SDS was a reliable, valid measure of the severity of cannabis use. A score ≥4 was indicative of cannabis dependence [15].

The primary outcome in the analysis was the number of days of cannabis use during the last 30 days before the assessments (this measure was related to the Addiction Severity Index) [16]. Complete abstinence was defined as 0 days of cannabis use for 30 days. The uses of alcohol, other substances, and nicotine during the prior week were measured with separate visual analog scales, which ranged from 0 (no use) to 10 (massive daily use) [17]. Motivation was assessed on a similar visual analogue scale, with a question about the importance of quitting cannabis use (0 = very little importance; 10 = highly important). The Hopkins Symptom Checklist (HSCL), 25-item version, was used to measure mental distress during the last 14 days [18]. Each of the 25 items was scored on a 4-point scale (range: 1–4). An average score was computed to obtain a global severity index of mental distress (GSI, range: 1–4), where higher scores indicated greater distress [18]. A GSI ≥ 1.75 indicated a clinical level of mental distress. The Outcome Rating Scale (ORS) was used to assess well-being. The scale had four visual analogue items on personal, interpersonal, social, and general well-being. Each item was scored from 0 (poor) to 10 (good) [19]. The four item scores were summed (range: 0–40), and a score <25 indicated scores expected in a clinical population. A ≥ 5-point improvement and scores above 25 were considered to indicate a clinically meaningful change [19].

To map the social network, respondents were simply asked to count the number of friends that they spent a substantial amount of time with, and to count how many did and did not currently use substances. The sense of coherence (SoC) refers to a person's ability to use existing and potential resources to combat stress, overcome resistance and difficulties, and promote health [20]. Antonovsky's SoC scale (SoC-29) measured the degree to which the subject felt that he or she had a sense of control over their own lives (manageability– 10 items), that life had meaning (8 items), and that their social life was understandable (comprehensibility– 11 items). Responses to items were scored on seven-point semantic differential scales. Scores were summed (range: 29–203), and higher scores indicated a stronger SoC [20].

## Ethical approval and consent to participate

The Regional Committee for Medical and Health Research Ethics in the South-East Health Region (REK 2012/1407) approved the study. All participants provided written informed consent before inclusion.

## Statistical analysis

Participant characteristics are presented descriptively. The paired sample t-test was used to examine changes in continuous variables from T0 to T1 and from T0 to T2. The McNemar test was used to examine changes in binomial variables. We performed multivariable logistic regression to explore whether essential severity and motivation variables were associated with abstinence from cannabis at follow-up. Due to sample size limitations at T2, this analysis was based on T0 and T1 data, and we could only include a few pre-specified variables (i.e., substance use and mental health severity, and the level of motivation at baseline). Results are presented as odds ratios with 95% confidence intervals (95% CIs). The significance level was set at $p < 0.05$. Statistical analyses were performed with IBM SPSS statistical software, version 25.0 (IBM Corporation, Armonk, NY).

## Results

The sociodemographic characteristics of the study respondents are shown in Table 2. The mean age of respondents was 25 years, most were male, living alone, and had completed only mandatory education ($\leq$10 years). Problematic cannabis use had lasted for a mean of 7 years, the mean debut age was 16 years old, and the mean SDS score was 8.8 (range: 0–15). Four out of ten respondents had some experience with synthetic cannabis, but only 5% had used synthetic cannabis on more than five occasions. The level of alcohol use and other substance use was also low at baseline; thus, cannabis was clearly the preferred substance (Table 2).

Seventy-six of 102 participants (75%) completed the CCP. An attrition analysis, based on the variables in Table 2, showed that those that dropped out were significantly younger (mean 22.5 vs. 26.1 years, p = 0.017) than those that completed the CCP (i.e., the included group). Otherwise, there were no socio-demographic differences between the two groups. In particular, there was no difference between groups in severity scores. Compared to the included group, the drop-out group had a mean SDS score of 8.5 (vs. 8.8, p = 0.629) and a mean HSCL-GSI score of 2.12 (vs. 2.18, p = 0.642) at baseline. The drop-out group had attended an average of 7.8 sessions, compared to 12.3 sessions in the included group. At the 3-month follow-up (T2), only 59 respondents (58%) could be reached when contact was attempted; of these, 53 were included, and 6 had dropped out at T1. Compared to those included in the analysis, those lost at T2 were younger, they had fewer friends without substance use problems at baseline (2.4 vs. 3.8 friends, p = 0.030), and they had more friends with substance use problems

**Table 2. Sociodemographic characteristics of study respondents.**

| Characteristics | *n* = 102 |
|---|---|
| Age, years | *25 (7)* |
| Sex, female | 26 (26) |
| Relationship: not married or living with a partner (n = 101) | 77 (76) |
| Education, years | *12.5 (1.9)* |
| Educational level | |
| Did not complete primary school | 3 (3) |
| Completed primary and secondary school (10 years of education) | 45 (44) |
| Completed high school (up to 13 years of education) | 44 (43) |
| Completed university ($\geq$ bachelor's degree) | 10 (10) |
| Occupation (*n* = 92) | |
| At least some income from work | 48 (52) |
| Number of days worked within the last 30 days | *11 (13)* |
| Cannabis use and severity | |
| Age at first-time use of cannabis, years | *16 (2)* |
| Duration of problematic cannabis use, years | *7 (6)* |
| Mean score on severity of dependence scale (SDS, scale 0–15) | *8.8 (2.8)* |
| Use of synthetic cannabis on any occasion (*n* = 98) | 39 (40) |
| Use of synthetic cannabis on >5 occasions (*n* = 98) | 5 (5) |
| Motivation: Importance of quitting (VAS–range: 0–10) | *9.1 (1.7)* |

Values are the number of participants (%) or the *mean (SD)*; VAS: visual analogue scale

at baseline (5.5 vs. 3.8, p = 0.027). Nevertheless, there were no differences in the perceived severity of cannabis use at baseline.

## Primary outcome (cannabis use)

When examining changes in cannabis use, there was a marked reduction in the number of days of cannabis use within the last 30 days. The mean reduction was −16.4 days (95% CI = −19.1 to 13.6, p <0.001) between T0 and T1, and the mean reduction was −12.6 days (95% CI = −16.4 to −8.8, p = 0.001) between T0 and T2 (Table 3). The proportion of participants that achieved complete abstinence from cannabis was 67% at T1 (51 of 76) and 37% at T2 (22

**Table 3. Self-reported changes in substance use among study respondents.**

| Variables | T0 | T1 | T2 | P-value[a] | P-value[a] |
|---|---|---|---|---|---|
| | n = 102 | n = 76 | n = 59 | T0–T1 | T0–T2 |
| Number of days cannabis was used within the last 30 days | *18.9 (10.2)* | *2.5 (6.3)* | *5.9 (9.4)* | <0.001 | <0.001 |
| Abstinence—no use of cannabis within the last 30 days–N (%) | 5 (5) | 51 (67) | 22 (37) | <0.001 | <0.001 |
| Alcohol use (VAS scale, 0–10) | *1.6 (1.7)* | *2.1 (2.3)* | *1.6 (2.0)* | 0.106 | 0.534 |
| Other drug use (VAS scale 0–10) | *0.2 (0.9)* | *0.1 (0.4)* | *0.2 (1.1)* | 0.156 | 0.857 |
| Nicotine use (VAS scale 0–10) | *7.3 (3.4)* | *6.7 (3.1)* | *7.1 (3.2)* | 0.123 | 0.283 |

Values are the number of participants (%) or the *mean (SD)*; T0 = baseline, T1 = after the treatment program (2 months), T2 = at the 3-month follow-up (i.e., 5 months after T0);

[a] P-values were obtained with the paired t-test; the T0 values used in the pairwise analysis may be slightly different, due to attrition at T1 and T2; the McNemar test was applied for repeated measures of binomial variables; VAS: visual analogue scale

of 59). With an intention-to-treat analysis, the proportions were 50% (51/102) and 22% (22/102) at T1 and T2, respectively (Table 3).

### Secondary outcomes (exploratory analyses)

The longitudinal data revealed that the reduction in cannabis use was not replaced by a significant increase in other substance use, including alcohol and other drugs. The use of nicotine was quite high, and it remained high throughout T2; i.e., no significant change was observed over time (Table 3). Consistent with the reduction in cannabis use, there was a substantial reduction in the mean mental distress score (HSCL-GSI) between T0 and T1 (−0.63 points, 95% CI = −0.76 to −0.50, p <0.001). This improvement was retained at T2 (Table 4). Accordingly, there was a large improvement in well-being. The mean ORS score increased between T0 and T1 by 8.2 points (95% CI = 5.9 to 10.4, p <0.001), and it increased between T0 and T2 by 6.4 points (95% CI = 3.7 to 9.1, p <0.001; Table 4). The mean SoC score also significantly improved from T0 to the other time-points, with a total mean increase of >18 points (Table 4). On the social level, there was a decrease in the mean number of friends with current substance use problems that the respondents spent time with. The mean number decreased from 4.5 to 2.8 at the other time-points. However, there was no significant increase in the drug-free social network (Table 4, number of friends without substance use).

### Factors associated with cannabis abstinence post-treatment

The multivariable regression analysis (Table 5) showed that the perceived severity of cannabis use at baseline was not associated with abstinence from cannabis use post-treatment (T1). However, mental distress was negatively associated with abstinence (OR = 0.28, 95% CI = 0.08 to 0.93, p = 0.037), and the perceived importance of quitting at baseline was positively associated with abstinence (OR = 1.04, 95% CI = 1.00 to 1.08, p = 0.031). Age and sex were not significantly associated with abstinence at T1.

## Discussion

The CCP had a high rate of retaining participants; 75% of participants completed the program. We observed a large reduction in cannabis use from baseline to T1 and T2, but less of a reduction at T2 than at T1. In parallel with the reduction in cannabis use, we observed

**Table 4. Self-reported changes in mental health, well-being, social network, and sense of coherence (SoC) among study respondents.**

| Variables | T0 (N = 102) | T1 (N = 76) | T2 (N = 59) | P-value[a] T0 –T1 | P-value[a] T0 –T2 |
|---|---|---|---|---|---|
| Mental distress (HSCL-GSI) | 2.16 (0.55) | 1.54 (0.45) | 1.51 (0.46) | <0.001 | <0.001 |
| Proportion of scores >cut-off for HSCL-GSI | 73 (72) | 22 (30) | 16 (27) | <0.001 | <0.001 |
| Well-being (ORS) | 22.4 (8.6) | 30.6 (7.8) | 29.1 (9.4) | <0.001 | <0.001 |
| Social network | | | | | |
| Number of friends without substance use | 3.2 (3.2) | 3.7 (2.8) | 4.3 (2.9) | 0.786 | 0.616 |
| Number of friends with substance use | 4.5 (3.8) | 2.8 (2.9) | 2.8 (3.0) | <0.001 | <0.032 |
| Sense of coherence | 115 (22) | 133 (21) | 135 (25) | <0.001 | <0.001 |

Values are the number of participants (%) or the *mean (SD)*; T0 = baseline, T1 = after the treatment program (2 months), T2 = at the 3-month follow-up (i.e., 5 months after T0);

[a] P-values were obtained with the paired t-test; T0 values used in the pairwise analysis may be slightly different, due to attrition at T1 and T2; the McNemar test was applied for repeated measures of binomial variables; GSI: Global Symptom Index; HSCL: Hopkins Symptom Check List; ORS: Outcome Rating Scale

**Table 5. Multivariable logistic regression results show associations between post-treatment cannabis abstinence (T1) and baseline predictors.**

| Variables | OR (95% CI) | P-value |
|---|---|---|
| Perceived severity of cannabis use (SDS) | 1.00 (0.77–1.30) | 0.987 |
| Mental distress (HSCL-25-GSI) | 0.28 (0.08–0.93) | 0.037 |
| Importance of quitting | 1.04 (1.00–1.08) | 0.031 |

The analysis was controlled for age and sex, which were not significant independent factors; GSI: Global Symptom Index; HSCL: Hopkins Symptom Check List; SDS: Severity of Dependence Scale

improvements in mental distress, well-being, and SoC measures. Participants reduced the number of friends that used substances, but did not significantly increase their drug-free social network.

The large reduction in cannabis use at T1 (average reduction: ~16 days per month) and at T2 (~13 days per month) was higher than the mean reduction in cannabis use (~6 days) reported previously with the controlled interventions included in the Cochrane review on psychosocial interventions for CUD [6]. Moreover, a high proportion (67%) of participants that completed the program had shown abstinence at the end of the program. The intention-to-treat analysis showed that half of the participants reported complete abstinence from cannabis use after the program. This proportion was slightly higher than that reported previously in a naturalistic Swedish CCP study, where 33 of 75 (44%) participants achieved abstinence after the intervention [9]. In contrast to observations in other CCP studies, the reduction we observed in cannabis use was not replaced by a significant increase in other substances, including alcohol and other drugs [9, 12]. The visual analogue scale was used clinically by the CCP therapists to thematize and follow up on the possibility that respondents might replace cannabis use with other substances. Taken together, these findings suggested that the CCP, as practiced in the Norwegian setting, was useful for assisting participants in reducing and/or discontinuing cannabis use. However, at T2, the attrition increased, cannabis use somewhat increased, and complete abstinence decreased. These findings indicated some uncertainty in the stability of reduced cannabis use.

The drop-out group was younger than the included group, and attrition at T2 was highest among the youngest participants. This finding corroborated the opinion of some of the therapists, who held that the youngest participants found it more difficult to continue and complete the program. This finding was consistent with previous SUD studies that observed a low follow-up rate among young participants [21]. Some studies observed that younger participants reported higher external pressure (e.g., family pressure) and less internal motivation to comply with SUD treatment; these factors might partly explain the high attrition among the youngest participants [22]. This finding warrants attention from a clinical perspective: clinicians should be aware of the extra challenges among younger individuals. The program might need to be strengthened for these participants; for example, by placing extra attention on the therapeutic alliance, by involving the participant's family, and/or by including mentor support or mentor guidance in an activity the youth is interested in. This extended approach would require resources and collaboration across municipal services.

Several studies have shown associations between cannabis use, elevated levels of mental distress, and poor well-being [4, 11]. In the present study, we observed simultaneous reductions in cannabis use and mental distress. This result indicated that, as a whole, our sample did not consist of individuals with mental distress as a primary problem, because the decrease in cannabis use facilitated relief from mental distress. This finding was consistent with findings from some controlled trials; for example, in the Marijuana Treatment Project study, reductions in

marijuana use at follow-up were associated with reductions in anxiety [23]. However, results from controlled studies have been mixed; thus, our findings contrasted with findings from other controlled trials. For example, in an Australian study, the mental distress outcome score was not better among those in specific cannabis interventions (two brief cognitive-behavioral therapy programs) than among those in a delayed-treatment control group, and the improvements observed were quite small [24]. However, in contrast to our cohort, the participants in that study had a pre-study mean mental distress score that was considerably below the clinical cut-off value; i.e., there might have been a floor effect.

The substantial increase in well-being that we observed in the present sample was consistent with our observations on mental distress. We observed a clinically relevant (>5-point) improvement in well-being, and the values were above the clinical cut-off values at both time-points after the intervention [19]. Similarly, there was a >18-point increase in SoC from baseline to the other time points. The therapeutic approach in CCP involves psychosocial coaching in the latter weeks; therefore, the improvement in SoC might have been due to the combination of abstinence or reduced consumption and an improvement in personal and social competences, consistent with Lundqvist's earlier findings [25]. However, the few previous cannabis treatment studies that included longitudinal SoC data have reported mixed results. In one study, after six weeks of abstinence in a specialized CCP treatment, the cannabis group improved their SoC score to the same level as that seen in a Swedish population sample [25]. In another study, Petrell et al. found equivalent improvement among adolescents [9]. However, an online intervention that was based on the CCP showed no improvement in SoC compared to an untreated control group [10]. In a previous study on workers with and without sick-leave from back pain, a 5-point difference between mean SoC scores was reported [26]. Therefore, the large increase in SoC scores observed in the present study indicated that CCP respondents must have experienced a substantial improvement. Nevertheless, our CCP respondents had a ~10-point lower score at T2 than the score reported by social work students in a previous study [27]. Thus, the CCP achieved a somewhat lower than optimal post-intervention SoC, and there remains room for improvement.

It has been shown that individuals with more support from a network of friends that used substances at treatment baseline were more likely to exhibit severe substance use after treatment [28]. In the present study, our respondents significantly reduced the numbers of friends that used substances, which may have contributed to the good results we observed. However, this change in the social network was not followed by a parallel increase in the size of their drug-free social network. Although one might question the realistic possibility of making a number of new friends within a timeframe of five months, this finding warrants attention from a clinical perspective. CCP workers should focus on the importance of engaging in positive activities that could facilitate the development of a positive social network. For example, a positive social network might be available via work or educational venues, sports, organized leisure activities, or support groups [21, 29].

A surprising finding from our regression analysis was that the perceived severity of cannabis use at baseline was not associated with abstinence from cannabis use post-treatment. This finding indicated that the severity of cannabis use was not predictive of the outcome. On the other hand, coexisting mental distress had a substantially negative impact on cannabis abstinence at T1, consistent with findings in the previous Cochrane review. Those authors noted that the three studies that included participants with severe psychiatric conditions did not report significant improvements in cannabis use at follow-up [6]. Taken together, those findings implied that patients with additional mental health symptomatology were at a disadvantage in accomplishing difficult life style changes, like cannabis cessation. Thus, this subgroup might need a more specialized treatment program with a greater focus on mental comorbidities.

## Methodological considerations

The main strength of this study was that the intervention examined was considered a low-threshold intervention, and it was performed at the lowest care level; the primary care level in the municipality.

This study also had some limitations. There was a relatively high attrition rate during follow-up, the follow-up period was relatively short, and we lacked biological data (e.g., urinary cannabinoid tests) to confirm the results. Moreover, the non-experimental design prevented us from drawing clear conclusions about the effectiveness of the intervention. However, it remains an ethical issue whether young adults should be randomized to an inactive control condition for extended periods, after problematic cannabis use has been identified, instead of immediately offering them the help they have sought [10]. Hence, we chose to conduct this naturalistic follow-up study, instead of allocating respondents to a delayed-treatment or a similar inactive condition. A potential way to address this dilemma might be to allocate some respondents to an alternative active condition, for example, an internet-based CCP. A digital intervention became available after the present study was completed [11]. Although the present participants were motivated to change, we suspect that the number of reported improvements was unlikely to have been the same if these participants had been allocated to a waiting list or to an inactive group that depended on self-change.

Another limitation was that the data were collected by professionals that were motivated to take part in the study, but not all professionals working with CCP participated. Moreover, the study did not include a method for monitoring recruitment. To the best of our knowledge, the participating professionals followed the procedures for inclusion in the study and invited all eligible subjects in their groups to participate.

## Conclusion

As practiced in some Norwegian municipalities, we found that the CCP was a valuable, low-threshold intervention with considerable potential for reducing young individuals' cannabis use in a primary care setting.

## Supporting information

**S1 File.**
(SAV)

**S2 File.**
(PDF)

## Author Contributions

**Conceptualization:** John-Kåre Vederhus, Malin Rørendal, Madelene Skårdal, Marianne Otterstad Næss, Øistein Kristensen.

**Formal analysis:** John-Kåre Vederhus.

**Project administration:** John-Kåre Vederhus.

**Supervision:** Thomas Clausen.

**Writing – original draft:** John-Kåre Vederhus, Malin Rørendal, Madelene Skårdal, Marianne Otterstad Næss, Thomas Clausen, Øistein Kristensen.

**Writing – review & editing:** John-Kåre Vederhus, Malin Rørendal, Madelene Skårdal, Marianne Otterstad Næss, Thomas Clausen, Øistein Kristensen.

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
