## [Decision Letter · Decision Letter 0]

16 Mar 2022

PONE-D-21-34465Successful outcomes with low–threshold intervention for cannabis use disorders in Norway - an observational studyPLOS ONE

Dear Dr. Verderhus,

Thank you for submitting your manuscript to PLOS ONE. After careful consideration, we feel that it has merit but does not fully meet PLOS ONE’s publication criteria as it currently stands. Therefore, we invite you to submit a revised version of the manuscript that addresses the points raised during the review process.

Please consider carefully all the reviewer´s comments about the manuscript

We look forward to receiving your revised manuscript.

Kind regards,

José J. López-Goñi

Academic Editor

PLOS ONE

Journal Requirements:

4. Ethics statement only appears at the end of the manuscript:

Your ethics statement should only appear in the Methods section of your manuscript. If your ethics statement is written in any section besides the Methods, please move it to the Methods section and delete it from any other section. Please ensure that your ethics statement is included in your manuscript, as the ethics statement entered into the online submission form will not be published alongside your manuscript. 

Additional Editor Comments:

Please consider carefully all the reviewer´s comments about the manuscript.

Reviewers' comments:

Reviewer's Responses to Questions

**Comments to the Author**

1. Is the manuscript technically sound, and do the data support the conclusions?

Reviewer #1: Partly

2. Has the statistical analysis been performed appropriately and rigorously? 

Reviewer #1: I Don't Know

3. Have the authors made all data underlying the findings in their manuscript fully available?

Reviewer #1: No

4. Is the manuscript presented in an intelligible fashion and written in standard English?

Reviewer #1: Yes

5. Review Comments to the Author

Reviewer #1: "Successful outcomes with low–threshold intervention for cannabis use disorders in Norway - an observational study."

1. The labels “low-threshold” and “low-intensity” are used to describe the intervention. Are these appropriate given this is a 15-session, therapist-led intervention including a family session, and with an emphasis on achieving abstinence from cannabis use?

2. The intervention is described as “a secondary prevention intervention” – however the participants already perceived their cannabis use as problematic in this study (and had SDS scores ≥4). Should this still be described as a preventative intervention in this case?

Methods:

3. Phase 1 of the intervention is described as preparing for physical withdrawal, and as a “medical” phase. Did this include any type of pharmacological intervention? If so, details of these would be needed.

Results:

4. The change in cannabis use from T0-T2 is unclear, in the main text: “the mean reduction was −12.6 days (95% CI = −16.4 to 8.8, p = 0.001)”. These confidence intervals would indicate a non-significant change. Is this an error?

5. I can’t see any data presented on how many participants were abstinent at T0. Given it is mentioned that: “They were advised to reduce or quit their cannabis use before starting the program, or during the first part of the program.”

How many were abstinent at the start of treatment? Why wasn’t baseline cannabis use adjusted for in the multivariable regression analysis?

6. The authors describe conducting “an attrition analysis” – which analysis was conducted?

7. The results should follow the outline given on the clinical trial registration. ‘Other drug use’ is presented with the primary outcome in the manuscript which is not appropriate as this was not listed as a primary or secondary outcome on the registration. Data on alcohol/other drug use could be moved to a section labelled “exploratory analysis”.

8. In Table 3 – what is the reason for writing “Ns” instead of the p-values that were >.05? These tables would also be improved by the addition of 95% confidence intervals. Table 3 needs to define what c refers to.

9. The text states “At the 3-month follow-up (T2), 59 respondents (58%) were contacted...”; If I am reading this correctly, why were nearly half of participants not contacted for this follow-up? Or does this number refer to the number of individuals that completed the follow-up?

10. What are “classical cannabis products” used by participants? The background suggests that resin may be partly responsible for increased admissions to treatment, do you have data on herbal/resin use in this sample?

11. When describing the sample characteristics, percentages or count would be better than “one in four”, “three of four” etc.

Discussion:

12. A brief sentence describing the intervention prior to describe the main findings about retention and cannabis use would improve readability of the discussion.

13. The reference provided to support not employing a waiting-list control (Sinadinovic, 2020) does not seem to noticeably support this suggestion. This point needs to be expanded, as this is a key criticism of the current investigation.

14. Further clarity and discussion are needed of the implications related to: “These professionals included only a proportion of those that were enrolled in the CCP in participating municipalities. We lacked information about the pre-study procedures; e.g., whether the professionals included all their eligible individuals in the study.”

6. PLOS authors have the option to publish the peer review history of their article (what does this mean?). If published, this will include your full peer review and any attached files.

Reviewer #1: No

---

## [Author Response · Author response to Decision Letter 0]

16 May 2022

Dear editor

Our sincere thanks for the valuable and constructive comments on our initial submission. 

We have responded to the reviewer and editorial comments in the "Response to Reviewer" letter. 

Yours Sincerely

John-Kåre Vederhus

---

## [Decision Letter · Decision Letter 1]

2 Jun 2022

Successful outcomes with low–threshold intervention for cannabis use disorders in Norway - an observational study

PONE-D-21-34465R1

Dear Dr. Vederhus,

We’re pleased to inform you that your manuscript has been judged scientifically suitable for publication and will be formally accepted for publication once it meets all outstanding technical requirements.

Kind regards,

José J. López-Goñi

Academic Editor

PLOS ONE

Additional Editor Comments (optional):

Reviewers' comments:

Reviewer's Responses to Questions

**Comments to the Author**

1. If the authors have adequately addressed your comments raised in a previous round of review and you feel that this manuscript is now acceptable for publication, you may indicate that here to bypass the “Comments to the Author” section, enter your conflict of interest statement in the “Confidential to Editor” section, and submit your "Accept" recommendation.

Reviewer #1: All comments have been addressed

2. Is the manuscript technically sound, and do the data support the conclusions?

Reviewer #1: Yes

3. Has the statistical analysis been performed appropriately and rigorously? 

Reviewer #1: Yes

4. Have the authors made all data underlying the findings in their manuscript fully available?

Reviewer #1: Yes

5. Is the manuscript presented in an intelligible fashion and written in standard English?

Reviewer #1: Yes

6. Review Comments to the Author

Reviewer #1: Thank you for addressing my comments and for sharing the data file. I would recommend sharing the data file as a CSV rather than an SPSS file as that will make it much more accessible.

7. PLOS authors have the option to publish the peer review history of their article (what does this mean?). If published, this will include your full peer review and any attached files.

Reviewer #1: No

---

## [Editor Report · Acceptance letter]

6 Jun 2022

PONE-D-21-34465R1 

Successful outcomes with low–threshold intervention for cannabis use disorders in Norway - an observational study 

Dear Dr. Vederhus:

I'm pleased to inform you that your manuscript has been deemed suitable for publication in PLOS ONE. Congratulations! Your manuscript is now with our production department. 

Kind regards, 

on behalf of

Dr. José J. López-Goñi 

Academic Editor

PLOS ONE